# An Exploratory In Vitro Microcomputed Tomographic Investigation of the Efficacy of Semicircular Apicoectomy Performed with Trephine Bur

Eszter Nagy [1], Brigitta Vőneki [1], Lívia Vásárhelyi [2], Imre Szenti [2], Márk Fráter [1], Ákos Kukovecz [2] and Márk Ádám Antal [1,3,*]

1 Department of Esthetic and Operative Dentistry, Faculty of Dentistry, University of Szeged, 6720 Szeged, Hungary; eszter.drnagy87@gmail.com (E.N.); brigitta.voneki@gmail.com (B.V.); meddentist.fm@gmail.com (M.F.)

2 Department of Applied and Environmental Chemistry, Interdisciplinary Excellence Centre, University of Szeged, 6720 Szeged, Hungary; vasarhelyil@hotmail.com (L.V.); szentiimre@gmail.com (I.S.); kakos@chem.u-szeged.hu (Á.K.)

3 College of Dental Medicine, University of Sharjah, Sharjah P.O. Box 27272, United Arab Emirates

* Correspondence: antal.mark@szte.hu

**Abstract:** Purpose: Recently, a novel approach to apicoectomy has emerged, involving the use of a hollow trephine with a surgical guide. This innovative technique creates a semicircular section, in contrast to the conventional straight cut (with a bur). The semicircular shape of this section raises questions about the method's effectiveness in eliminating supernumerary canals (SNCs), which are auxiliary canals alongside the main root canal. These canals pose a risk of further infection if not thoroughly eliminated. The aim of this exploratory study was to assess the efficacy of the proposed method in the removal of SNCs. Methods: A sample of 200 extracted single-rooted human teeth was rigorously narrowed down by multiple steps to 33 specimens that contained SNCs in the apical third. These specimens underwent apical semicircular sectioning, removing the apical 3 mm of the root. The specimens then underwent repeated CT scanning. For samples with residual SNCs, the section was transformed into a straight section and another CT scan was performed. Results: The semicircular section eliminated all SNCs in 94% of the specimens, and it eliminated 97.3% of the SNCs in all specimens. The straight-line sectioning eliminated 98.6% of all SNCs and eliminated all SNCs in 96.97% of the SNC-containing teeth. Conclusions: The efficacy of semicircular apicoectomy performed with a trephine appears to be comparable ($\chi^2 = 1.00$, df = 1, and $p = 0.317$) to that of the conventional straight apicoectomy performed with a bur as reported in the literature.

**Keywords:** apicoectomy; trephine; endodontic surgery; surgical techniques

## 1. Introduction

Conventional endodontic treatment is based on the mechanical removal of the infected pulp tissue, chemical disinfection, and root canal filling. The success rate can reach 98% according to the literature [1]. However, the success rate of orthograde treatment is highly determined by the anatomical knowledge [2,3], and even this cannot provide complete success on its own. Complex apical anatomy, e.g., ramifications, $^132^{1-2-1}$ [4], or Type VII [5,6] configurations, can make sufficient orthograde cleaning impossible. In most of these cases, the residual periapical inflammation cannot be resolved by re-treatment either. So, in these cases, the retrograde approach, apicoectomy, is the proper treatment of choice [7].

Apicoectomy is the surgical removal of the apical portion of the root to eliminate any remaining reservoir of infection (lateral canals or ramifications that cannot be instrumented from the orthograde direction). To effectively remove these structures, it is advisable to

excise the apical 2 to 3 mm of the final third of the apex [8,9]. According to the modern principles, the section must be performed perpendicularly to the root axis to minimize the chance of bacterial leakage through the dentinal tubuli [10]. Following these principles, the elimination rates may be as high as 93 to 98% [11]. The perpendicular technique was a great innovation compared to the previously applied 45° technique with 60 to 70% elimination rates [12]. However, as the operation is conventionally carried out with burs or a piezoelectric device [8,12] and without any physical guidance, it is still an invasive procedure with prolonged and often painful healing. Therefore, these procedures should tend towards minimal invasiveness, which appears to be addressed most efficiently by increasing the accuracy [13,14].

The first evidence that CAD/CAM guides can be used to increase the accuracy of apicoectomy was published by Pinsky and colleagues in 2007 [15]. The in vivo application of this idea only appeared 10 years later, due to the steep development of stereolithographic techniques. At first, only the localization of the apex was guided to help the osteotomy performed by Liu et al. [16]. Another approach was to use a 3D-printed soft tissue retractor during the apicoectomy procedure [17]. The implementation of 3D implant planning software into endodontic surgical procedures made the optimization of depth, angle, and localization possible [18–20]. The very first steps of the evolution of this technique were focusing on a non-guiding template, only to be an optical reference prior to or during the free-hand apicoectomy. Giacomino and co-workers [21] were the first to use a trephine bur to perform a fully guided osteotomy, apicoectomy, and biopsy at the same time. Since then, this method has been discussed in various studies [22–30], mainly case presentations. The accuracy and efficacy of this method have not been examined so far. The trephine creates a semicircular section [23,31]. Given that the resected surface does not form a perpendicular flat line with the root axis, the remaining concave surface may harbor lateral canals or other potential bacterial reservoirs (Figure 1). Consequently, a situation arises where, despite the potential superior accuracy and precision of a guided trephine in comparison to conventional approaches, clinical effectiveness might be compromised due to the aforementioned concern. As this method gains popularity, there is a pressing need for dedicated preclinical investigations to elucidate and address this potential issue.

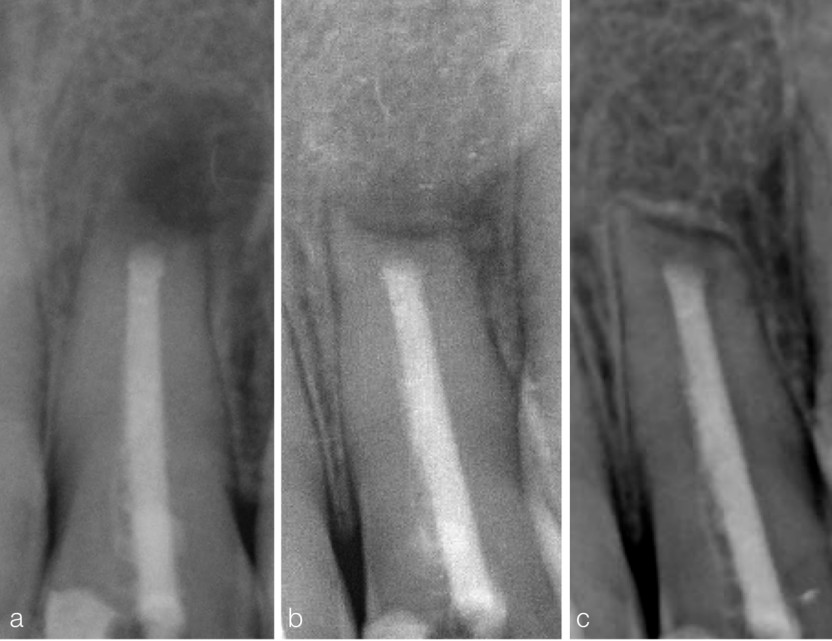

**Figure 1.** Clinical (periapical X-ray) images of guided endodontic surgery performed with a trephine bur. (**a**) Right after surgery; (**b**) at the six-month follow-up; (**c**) at the three-year follow-up.

The objective of this investigation was to undertake a preliminary assessment of the efficacy of trephine apicoectomy as an innovative technique, focusing on its capacity to effectively eradicate supernumerary canals. As a null hypothesis, we hypothesized that the efficacy of semicircular sectioning performed with a trephine would be comparable to that of the conventional straight section performed with a bur.

## 2. Materials and Methods

This study conformed to the Declaration of Helsinki in all respects. The protocol was approved by the Regional and Institutional Committee of Science and Research Ethics, University of Szeged, Hungary (Approval No.: RKEB 52/2018-SZTE).

This study was carried out on extracted single-rooted human incisors and canines. The initial sample consisted of 200 specimens extracted at the Faculty of Dentistry, University of Szeged. Only teeth with a single, structurally intact root were eligible for inclusion. Exclusion criteria encompassed previous apical resection, root resorption, double roots, and the presence of metallic restorations. The inclusion and exclusion criteria were assessed through visual inspection. The remaining specimens underwent a progressive selection process as follows: The teeth were first visually examined by transillumination (DiaLux 2300L Kavo, Germany) under 12.5× magnification (Zeiss OPMI Pico, Carl Zeiss AG Germany). All specimens with no visible accessory canals (as judged by the presence or absence of foramens) or with visible accessory canals outside the apical third were excluded. The four-eye principle was adhered to during the inclusion and exclusion process of the teeth, as well as in the initial visual selection step. This meant that any single tooth had to be agreed upon by two investigators, ensuring alignment with the specified criteria. The remaining teeth were radiologically tested with a multiscale X-ray microtomograph (Bruker Skyscan 2211, Bruker, Belgium). First, a prescan was performed in search of radiologically identifiable lateral canals. Teeth with no detectable ramifications or lateral canals were excluded. Specimens that passed this scan underwent a full scan at 2.5 μm pixel resolution, using a 1 mm Al filter, with the following source parameters: 110 kV tube voltage, 300 μA tube current, and 600 ms exposition time. The final study sample consisted of teeth in which lateral canals and/or ramifications could be identified with this method at the apical 3 mm portion of the apex (n = 33) (Figure 2).

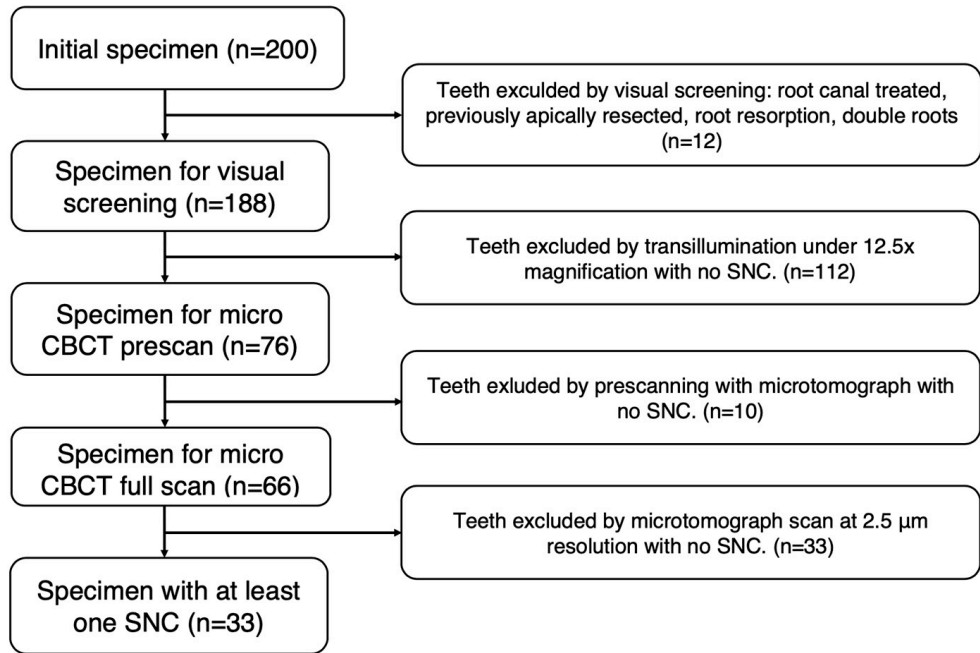

**Figure 2.** Flow diagram of screening and selection process of included teeth.

The 'Apicoectomy' itself was performed as previously described by Antal et al. [32], with hollow trephine burs of a 4.21 mm outer diameter (Hager & Meisinger, Neuss, Germany) at 800 rpm with a surgical motor (Implantmed W&H Dentalwerk, Bürmoos, Austria). Copious water cooling (0.9% saline, 105 mL/min) was applied. The axis of the section was perpendicular to the axis of the root. The apical 3 mm was removed according to the following steps: (1) a line, measured at 3 mm from the tip of the apex, was marked on the root surface with a 0.4 mm point black permanent marker (Stabilo OHPen S, Stabilo, Germany); (2) with the help of Krampon pliers, the tooth was stabilized over a metal sheet with its palatal surface positioned coronally from the marked 3 mm line, to prevent any interference during the sectioning; (3) the trephine bur was held with its uppermost point of outer curvature touching the pre-marked line and perpendicular to the axis of the root (Figure 3).

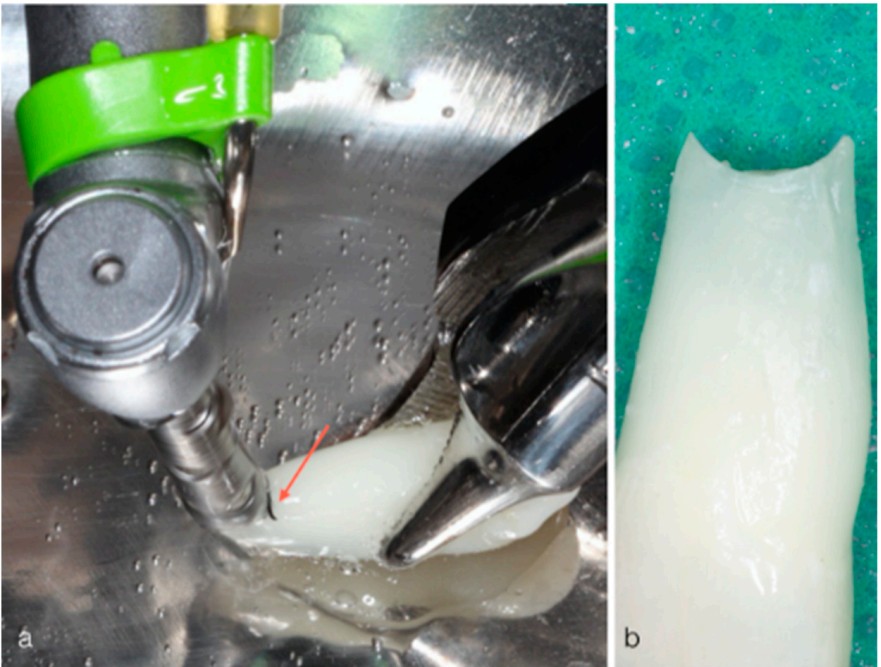

**Figure 3.** (**a**) Removal of the apex tip at 3 mm; (**b**) the characteristic concave outcome.

The apicoectomy was followed by micro-CT scanning, with the previously mentioned settings, to detect possible remaining lateral accessory canals and ramifications (SuperNumeraryCanal, SNC, Figure 4).

Teeth with the presence of the above-mentioned anatomical structures were detected. After this, the section line of each tooth was modified into a straight one along the 3 mm marked line, according to the principles of modern apicoectomy, in order to eliminate all tooth material between the semicircular cut surface and the pre-marked line. The straight-line cut was performed with Lindemann burs (Hager &Meisinger, Germany) at 1600 rpm. One more CT scan was performed to verify that the second section had eliminated the aforementioned all possible residual canal alterations. All sectionings were performed by the same investigator (EN). The method is summarized in Figure 5.

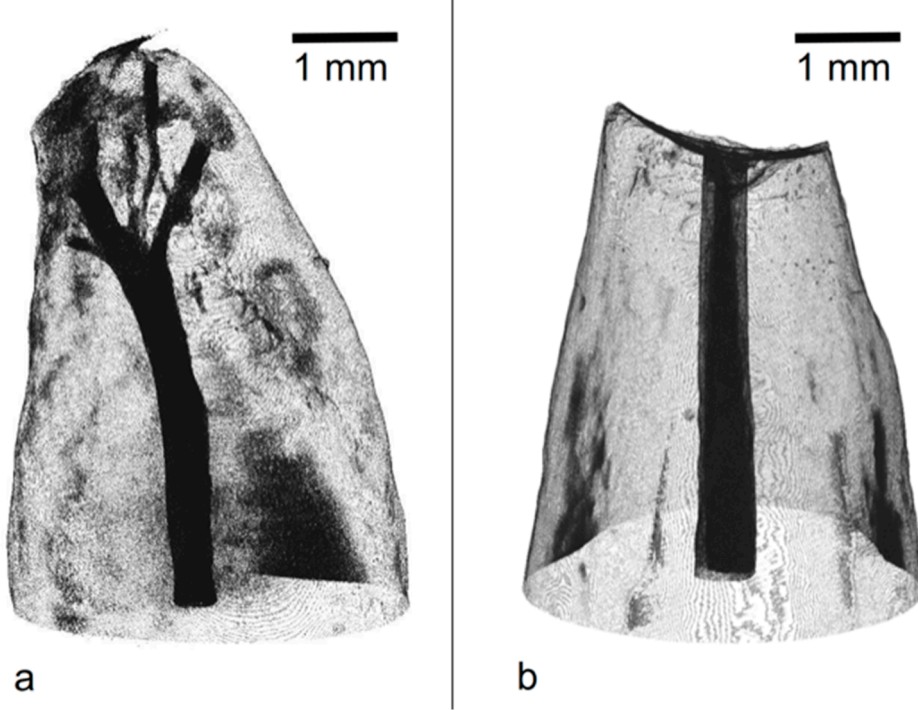

**Figure 4.** CT images before (**a**) and after (**b**) the semicircular section of the apex.

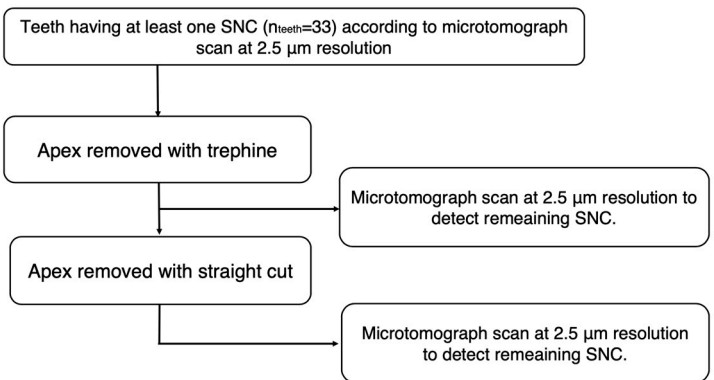

**Figure 5.** Flow diagram of apex removal and repeated CBCT scans.

In this study, SNCs were defined as any CT-identifiable additional canal (lateral canal or ramification) beside the main root canal. The number of teeth containing SNCs and the number of SNCs (by counting them in the CT images) were both documented. The complete removal of SNCs with semicircular section was determined in percentage, just like the number of straight cuts necessary to remove the SNCs remaining after the original sectioning method. Calculations were performed for the exact number of SNCs for the latter. The number of SNCs was listed by tooth type (central incisor, lateral incisor, or canine). Calculations were performed on the mean number of SNCs by tooth type as well.

The efficiency of the applied methods (semicircular sectioning with a trephine bur or conventional straight cut), defined as the ability to eliminate SNCs, was expressed in (a) the percentage of specimens in which the method eliminated all SNCs; (b) the percentage of the eliminated SNCs in the total number observed in all study specimens; and (c) a comparison of the efficacy of the 2 types of resection. For the latter comparison, beyond the descriptive analysis (percentages), the McNemar test was used, as this is a test for paired samples, and it can be used with relatively small sample sizes as well.

## 3. Results

A total of 76 specimens underwent radiological imaging with a micro-CT prescan to verify the presence of supernumerary canals (as described in Figure 2). The remaining 66 specimens underwent a full micro-CT scan. By micro-CT, the presence of SNCs in the apical 3 mms of the root could be verified in only 33 teeth, which formed the final study sample. The frequency of SNCs in the entire 188-specimen sample was 17.6%. Broken down by tooth type and referring to the entire 188-specimen sample, 24.6% of the central incisors, 9.6% of the lateral incisors, and 20.7% of the canines contained SNCs. For a summary of the selection procedure, see Table 1.

**Table 1.** Specimen numbers after each step of the progressive selection method. The sequence was to verify the presence of ramifications/lateral canals. INIT: initial specimen count (full sample); VIS: visual screening; TI: transillumination; PS: prescan; FS: full scan (see text).

|  | INIT | VIS | TI | PS | FS |
|---|---|---|---|---|---|
| Teeth (N) | 200 | 188 | 76 | 66 | 33 |
| Percentage (%) | NA | 100% | 40.4% | 35.1% | 17.6% |

The final study sample (N = 33) contained/consisted of 14 central incisors (42.4%), 7 lateral incisors (21.2%), and 12 canines (36.4%). The mean number of SNCs was 1.86 ($\pm$0.77) in central incisors, 1.86 ($\pm$1.06) in lateral incisors, and 2.90 ($\pm$2.30) in canines. The most frequent SNC numbers were one or two (in 23 cases, 69.7%). The highest detected number of supernumerary canals was eight (in one canine, 3%). The 33 teeth had 72 SNCs in all. Descriptive statistics of the study sample are given in Table 2.

**Table 2.** Specimen and supernumerary canal (SNC) counts and the corresponding percentages by tooth type for the study sample (N = 33).

| SNC Count | Central Incisor (N = 14) | | | | Lateral Incisor (N = 7) | | | | Canine (N = 12) | | | | Total (N = 33) | | | |
|---|---|---|---|---|---|---|---|---|---|---|---|---|---|---|---|---|
| | $N_{TEETH}$ | % | $N_{SNC}$ | % | $N_{TEETH}$ | % | $N_{SNC}$ | % | $N_{TEETH}$ | % | $N_{SNC}$ | % | $N_{TEETH}$ | % | $N_{SNC}$ | % |
| 1 | 5 | 35.7 | 5 | 19.2 | 3 | 42.9 | 3 | 23.1 | 6 | 50.0 | 6 | 18.2 | 14 | 42.4 | 14 | 19.4 |
| 2 | 6 | 42.9 | 12 | 46.2 | 3 | 42.9 | 6 | 46.2 | 0 | 0.0 | 0 | 0.0 | 9 | 27.3 | 18 | 25.0 |
| 3 | 3 | 21.4 | 9 | 34.6 | 0 | 0.0 | 0 | 0.0 | 3 | 25.0 | 9 | 27.3 | 6 | 18.2 | 18 | 25.0 |
| 4 | 0 | 0.0 | 0 | 0.0 | 1 | 14.3 | 4 | 30.8 | 1 | 8.3 | 4 | 12.1 | 2 | 6.1 | 8 | 11.1 |
| 5 | 0 | 0.0 | 0 | 0.0 | 0 | 0.0 | 0 | 0.0 | 0 | 0.0 | 0 | 0.0 | 0 | 0.0 | 0 | 0.0 |
| 6 | 0 | 0.0 | 0 | 0.0 | 0 | 0.0 | 0 | 0.0 | 1 | 8.3 | 6 | 18.2 | 1 | 3.0 | 6 | 8.3 |
| 7 | 0 | 0.0 | 0 | 0.0 | 0 | 0.0 | 0 | 0.0 | 0 | 0.0 | 0 | 0.0 | 0 | 0.0 | 0 | 0.0 |
| 8 | 0 | 0.0 | 0 | 0.0 | 0 | 0.0 | 0 | 0.0 | 1 | 8.3 | 8 | 24.2 | 1 | 3.0 | 8 | 11.1 |
| Total | 14 | 100 | 26 | 100 | 7 | 100 | 13 | 100 | 12 | 100 | 33 | 100 | 33 | 100 | 72 | 100 |

CT scans after the semicircular sections revealed two remaining SNCs in two different teeth altogether. After this, all specimens underwent a corrective (straight-line) section. After the straight-line sectioning, a residual SNC remained within one root only. Interestingly, in this specific sample, the SNC was located exactly at the level of the straight cut, slightly coronally—so that even with the classical straight resection, it persisted. Thus, the semicircular section eliminated all SNCs in 94% of the study samples, and it eliminated 97.3% of the total number of SNCs in all study samples. The success rate of the straight-line cut regarding the total number of SNCs in all study samples was 98.6% and 96.97% for the SNC-containing teeth.

The changing of the SNC numbers during the process is shown in Table 3, for an easier understanding of our results.

**Table 3.** Initial SNC numbers and their change after the semicircular and the straight-line apicoectomy ($\chi^2 = 1.00$, df = 1, and $p = 0.317$, according to the McNemar test).

| | Number of Teeth with SNC | Number of SNC | Number of Teeth without SNC | Eliminated SNC |
|---|---|---|---|---|
| Teeth included in the last phase | 33 | 72 | NA | NA |
| Microtomograph results after semi-curcular cut | 2 | 2 | 31 | 70 |
| Microtomograph results after straight cut | 1 | 1 | 32 | 71 |

To compare the efficiency of the circular and straight approaches in terms of how many teeth still contained SNCs after their application (regardless of the number of SNCs), the McNemar test was used. After the application of the semicircular trephine, 2 of the 33 teeth contained SNCs, while after the application of the semicircular trephine, 1 of the 33 teeth contained SNCs. The McNemar test yielded a non-significant result ($\chi^2 = 1.00$, df = 1, and $p = 0.317$), indicating no significant difference in the occurrence of SNCs between the circular and straight approaches. In other words, had the straight cut not been applied after the semicircular cut, the result would not have been statistically inferior.

## 4. Discussion

In this study, we sought to determine in vitro if the efficacy of semicircular apicoectomy performed with a trephine is comparable to that of the conventional straight apicoectomy performed with a bur. We embarked on this exploratory investigation because this method is gaining popularity in practice, yet lacks preclinical evidence to substantiate its non-inferiority compared to the conventional approach to apicoectomy in terms of eliminating SNCs. We hypothesized that the efficacy of semicircular sectioning performed with a trephine would be comparable to that of the conventional straight section performed with a bur.

According to our results, the hypothesis was confirmed. In our study, the proposed semicircular method eliminated all SNCs in 94% of the study specimens. Also, when looking at the total number of SNCs, it eliminated 97.3% of SNCs in all study specimens. The given percentages correspond almost exactly to those in the literature regarding the conventional straight-cut approach [11]. The comparison of the two methods did not show any significant difference regarding the complete removal of the SNCs from the apical 3 mm ($\chi^2 = 1.00$, df = 1, and $p = 0.317$). The treatment of the 2 teeth (6.1%) out of the 33 specimens with residual canals failed due to reasons that were not related to the method itself. In one sample, the problematic SNC was located coronally to the section line, so that it would have been impossible to remove it by following the conventional straight-line protocol. The presence of the other SNC was caused by operator error: asymmetric sectioning was performed as a result of misaligning the trephine bur to the midline of the root axis (Figure 6). As a result, only one side of the apex was eliminated properly, while a thick, triangular tooth structure remained on the contralateral side, involving the residual SNC. With the precise positioning of the trephine bur, these types of errors can be avoided, and the efficacy of the semicircular technique can be even more similar to the straight-line cut. In live surgeries, with the help of accurate digital planning followed by guided apicoectomy with the help of a 3D-printed surgical template, this operator error can easily be prevented. On the other hand, a misaligned apicoectomy can be intentional in certain cases, as Tavares and colleagues pointed out, to preserve adjacent anatomical structures [33].

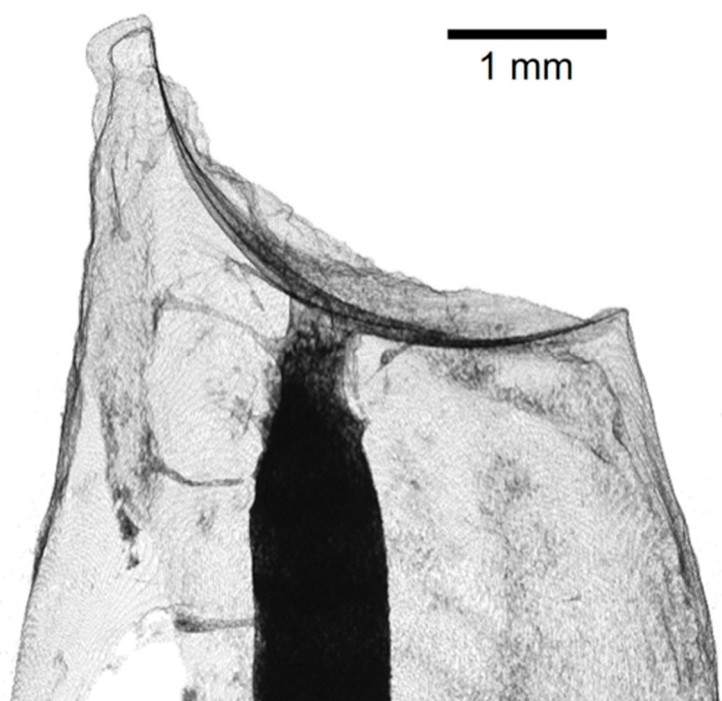

**Figure 6.** Residual SNCs as a result of operator error: the trephine was misaligned to the right, and this resulted in an asymmetric section.

As in most of the in vitro studies, the given optimal circumstances and precise following of the guideline can provide the result of an optimal procedure, resecting the apex exactly at 3 mm from the apex of the root [34–36]. This presupposes considerable accuracy. An in vivo apicoectomy with a trephine bur, in most of the cases, is a guided procedure, where a static surgical guide is providing stable circumstances. In the field of guided implant dentistry, guided surgery is considered to yield excellent accuracy [37], so that, theoretically, the 3D-printed guide based on the surgical plan and patients' individual anatomy should provide good enough accuracy to achieve the detected high SNC elimination percentages. As this type of guided technique is relatively new in surgical endodontics, hardly any empirical evidence is known to draw firm conclusions from. Other studies of ours imply that static guidance with trephine bur apicoectomy does provide a comparable accuracy to the accuracy of static guide-assisted implant placement documented in the literature [23,32]. It is not unrealistic to assume that in practice, trephine apicoectomy with static guidance can achieve the accuracy needed for the high efficiency we observed; however, further investigation is clearly necessary.

To have comparable results, our sample needed to reflect the incidence of SNCs described in the literature. According to Vertucci [6], the incidence of SNCs in maxillary central incisors was 24%, 26% in lateral incisors, and 30% in canines. Adorno and colleagues detected comparable ratios of SNCs in their study: 46% of central incisors, 29% of lateral incisors, and 38% of canines [38]. Kasahara and co-workers showed that SNCs were detectable in over 60% of the examined teeth [39]. In the present study, we found SNCs in 24.6% of the central incisors, 9.6% of the lateral incisors, and 20.6% of the canines. Interestingly, in our study, the incidence of SNCs was lower in the case of lateral incisors compared to the data reported in the literature [40–42]. This is a relative limitation of the study, resulting from the smaller number of lateral incisors in the original group of specimens. Thus, conclusions should not be generalized to this tooth type based on our results.

The authors consider it an interesting finding that within the limitations of our study, the mean number of SNCs in canines ($2.75 \pm 2.30$) was higher than in central incisors ($1.86 \pm 0.77$) or lateral incisors ($1.86 \pm 1.06$). Furthermore, only in canines did we observe

figures >4 SNCs (up to 8). Whether this finding indicates a unique characteristic of canines is too early to tell, as the (scarce) literature focusing on canine root canal anatomy concentrates almost exclusively on morphology, not the occurrence of SNCs in the apical area [43,44].

Finally, readers are encouraged to interpret the outcomes of this study in light of its inherent strengths and limitations. To the best of our knowledge, this marks the first endeavor to assess the efficacy of trephine apicoectomy in terms of supernumerary canal (SNC) elimination. The significance of this inquiry is underscored by the procedure's increasing prevalence, despite a dearth of preclinical data substantiating its non-inferiority to the conventional approach. Another notable strength lies in this study's utilization of micro-CT, enabling the precise quantification of SNCs, including the smallest ones. However, it is essential to acknowledge that this study is fundamentally exploratory, characterized by a relatively modest sample size, which is a definite limitation. Additionally, this study's in vitro design curtails the direct transferability of results to clinical contexts.

## 5. Conclusions

Within the limitations of this study, we conclude that the efficacy of semicircular apicoectomy performed with a trephine bur is comparable to that of the conventional, straight apicoectomy performed with a bur.

**Author Contributions:** E.N.: conceptualization, methodology, and writing—reviewing. B.V.: investigation, data curation, and writing—reviewing. L.V.: investigation, visualization, and writing—reviewing. I.S.: investigation, data curation, and writing—reviewing. M.F.: methodology, supervision, and writing—reviewing. Á.K.: methodology, supervision, and writing—reviewing. M.Á.A.: conceptualization, supervision, and writing—original draft preparation. All authors have read and agreed to the published version of the manuscript.

**Funding:** This research received no external funding.

**Institutional Review Board Statement:** This study was conducted in accordance with the Declaration of Helsinki and approved by the Regional and Institutional Committee of Science and Research Ethics, University of Szeged, Hungary (Approval No.: RKEB 52/2018-SZTE).

**Informed Consent Statement:** Informed consent was gathered from all patients as part of routine documentation at clinical procedures.

**Data Availability Statement:** Data are available from the corresponding author upon reasonable request.

**Conflicts of Interest:** The authors declare no conflict of interest.

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
