# Peer review of "An Exploratory In Vitro Microcomputed Tomographic Investigation of the Efficacy of Semicircular Apicoectomy Performed with Trephine Bur"

_applsci, doi:10.3390/app13169431_

Round 1

Reviewer 1 Report

Dear authors,

This study has an interesting topic. However, some issues need to be changed before the paper is approved for publication:

  1. Methods

Line 146-150 In paragraph about statistical analysis, you wrote: „Efficiency of the applied methods (semicircular sectioning with a trephine bur or con- 146 ventional straight cut), defined as the ability to eliminate SNCs, was expressed in a) the 147 percentage of specimens in which the method eliminated all SNCs and b) the percentage 148 of the eliminated SNCs in the total number observed in all study specimens, c) comparison 149 of the efficacy of the 2 types of resection in percentage.” Please, put here that you used the McNemar test, as you wrote in the results

  1. Results 

Please, put the results of The McNemar test in Table 3. (df, p, χ²).

  1. Discussion

Line 233-235 You wrote: “As in most of the in vitro studies, the given optimal circumstances and precise following of the guideline can provide the result of an optimal procedure, resecting the apex exactly at 3mm from the apex of the root.” Please, put here the references of these studies.

Line 254-255: You wrote: Interestingly, in our study the incidence of SNCs was lower in case of lateral incisors compared to the data reported in the literature.” Please, put here the references of these studies.

It is necessary to find more studies and compare the results with yours.

- What are the strenghts of your study? Please, describe it in a few sentences.

Minor editing of English language required.

Author Response

Dear Reviewer,

Please see our detailed responses below:

  1. Methods

Line 146-150 In paragraph about statistical analysis, you wrote: „Efficiency of the applied methods (semicircular sectioning with a trephine bur or con- 146 ventional straight cut), defined as the ability to eliminate SNCs, was expressed in a) the 147 percentage of specimens in which the method eliminated all SNCs and b) the percentage 148 of the eliminated SNCs in the total number observed in all study specimens, c) comparison 149 of the efficacy of the 2 types of resection in percentage.” Please, put here that you used the McNemar test, as you wrote in the results

R: Information added as requested

  1. Results 

Please, put the results of The McNemar test in Table 3. (df, p, χ²).

R: The results have been added to the caption of the table.

  1. Discussion

Line 233-235 You wrote: “As in most of the in vitro studies, the given optimal circumstances and precise following of the guideline can provide the result of an optimal procedure, resecting the apex exactly at 3mm from the apex of the root.” Please, put here the references of these studies.

R: The references have been added

Line 254-255: You wrote: Interestingly, in our study the incidence of SNCs was lower in case of lateral incisors compared to the data reported in the literature.” Please, put here the references of these studies.It is necessary to find more studies and compare the results with yours.

R: The references have been added

What are the strenghts of your study? Please, describe it in a few sentences.

R:  A strengths and weaknesses paragraph has been added to the end of the manuscript.

Reviewer 2 Report

Authors attempted an in vitro microcomputed tomographic investigation of the efficacy of guided semicircular apicoectomy performed with trephine bur. There are few comments based on the current manuscript.  

1) The abstract and introduction section has been presented well which has addressed the topic satisfactorily.

2)  The tables have been presented well with necessary details. However, on figure 1) Based on the present X-ray, the evolution of the quality of endodontic treatment is difficult to understand. It looks like the diameter of the filler is much smaller at the 3-year control compared to post-op. Perhaps, it would be wise to put better quality images or to change the case.  

4) The language and grammar of the manuscript is scientifically appropriate for the journal.  

5) The conclusion part is presented well.

Author Response

Dear Reviewer,

Thank you for the time you spent reviewing our manuscript and thanks for the comment regarding the 3-year control X-ray. As requested, the 3- year control x-ray in figure 3 has been replaced with an image of better quality.

Reviewer 3 Report

The manuscript lacked originality and failed to interest the reader. In addition, the manuscript is of poor quality.

The views discussed in this article are of little significance.

Some images are unnecessary to include in the article. For example, Figure 4.

Moderate editing in English is required.

Author Response

Thank you for dedicating your time to review our manuscript. While we value your perspective, we have done our best to enhance the manuscript based on suggestions from the other reviewers. We would like to emphasize that Figure 4 in particular displays the image depicting the semi-circular cut, a modification requested by another reviewer.

Reviewer 4 Report

I congratulate myself with the authors for the clear research presented. The content is interesting for the readers under my point of view. I would suggest to delete the word "guided" from the title since the apicoectomies in this study were not performed using guides even though the technique itself is aplicable for guided surgeries. 

Author Response

Dear Reviewer,

 Thank you for the time you spent reviewing our manuscript and thanks for pointing out the inaccuracy of the title. You are right indeed, it has been deleted.

Reviewer 5 Report

Abstract

The objective of this article is not clear. The purpose of this study was to examine the efficacy of the proposed method regarding SNC elimination. Please give the proposed method means in the objective of the abstract.

At the end of the Introduction, it is better to provide a clear objective.

Method.

Please provide details on the sample size calculation. 

It is better to provide inclusion and exclusion criteria for the selected teeth.

Results

It would have been better to add images of the root after apicoectomy.

Discussion

Add limitations of this research.

Minor correction is required.

Author Response

Dear Reviewer,

 Please see our detailed responses below:

Abstract

The objective of this article is not clear. The purpose of this study was to examine the efficacy of the proposed method regarding SNC elimination. Please give the proposed method means in the objective of the abstract.

R: The purpose section has been rephrased to better express the idea that the purpose of this exploratory study was to assess the trephine-based technique’s ability to eliminate supernumerary canals. It has also been clarified that this is a relatively novel approach that differs from the conventional approach (hence the study), and the importance of SNC elimination is also briefly explained. We believe this throws enough light on the purpose of the study: to have an initial look at the efficiency of a novel method in terms of a specific parameter. 

At the end of the Introduction, it is better to provide a clear objective.

R: We have added this sentence: „The objective of this investigation was to undertake a preliminary assessment of the efficacy of trephine apicoectomy as an innovative technique, focusing on its capacity to effectively eradicate supernumerary canals.” – in retrospect, this has improved the quality of the introduction indeed, so thanks for the suggestion.

Method.

Please provide details on the sample size calculation. 

R: In the initial iteration of the manuscript, our portrayal of the study as exploratory may not have been sufficiently explicit. Consequently, our approach was not geared towards achieving statistical power; rather, our focus was directed towards discerning marked differences between the two methods. The rationale behind selecting the McNemar test was twofold: its suitability for datasets akin to ours—characterized by dependent samples arranged in a 2 by 2 framework with binary outcomes—and its applicability even with modest sample sizes. The study's sample size, undeniably, presents a limitation, a facet we have now unequivocally articulated in the limitations section of this revised version. By accentuating the exploratory essence of the study, we aim to empower readers to interpret the results within their contextual significance.

It is better to provide inclusion and exclusion criteria for the selected teeth.

R: In the revised version, we mention the inclusion and exclusion criteria explicitly.

Results

It would have been better to add images of the root after apicoectomy.

R: Such images are shown in Figures 1 (x-ray), 3b (an actual root) and 4b (micro-CT).

Discussion

Add limitations of this research.

R: Limitations have been added to the end of the manuscript.

Reviewer 6 Report

Hi, 

The study has performed okay.

I have made a few comments where I felt some of the point were unclear. Please review.

Thank you

Moderate. Need some rephrasing of the sentences.

Author Response

Dear Reviewer,

Thank you for reviewing our manuscript. We believe we have addressed the points you raised properly. The modifications you requested have been made. A few questions we would like to address in more detail:

Was any dye used to detect the accessory canals? (p3)

No, dye was not utilized in our study, which is why it was not referenced in the manuscript. While it remains a possibility that this approach may have led to the potential exclusion of a few teeth harboring subtle miniature SNCs, whose orifices were not readily visible or easily discernible under the microscope, such an occurrence is highly improbable. Importantly, the omission of dye did not compromise the primary objective of our study—to evaluate the efficacy of the two methods in eliminating SNCs—since the quantification of SNCs was accurately conducted based on the CT images.

Please rearrange the discussion and talk more on the different methods used for removal of accessory canal at the apical 3mm of the root (p9).

The truth is that the range of methods is very limited, there is not much to discuss. Beside this novel guided trephine approach, it is possible to section the apex with a bur or a piezoelectric device. We have mentioned these in the Introduction (with references), but there is really not much more to say about them than what we already mentioned there: they are more invasive than the trephine approach and, as they come without a guide, it is easier to miss the 3 mm target with them. 

Please discuss clinical limitation of the study and what are the pro and cons of a semicircular method for further seal (p 9).

In the current rendition, we explicitly acknowledge that the in vitro nature of this study imposes constraints on the direct applicability of the results to clinical scenarios. Moreover, an inclusion has been made in the introduction to address the contentious aspect: while the semicircular method holds the promise of enhanced accuracy (pro), there exists the potential for it to be rendered clinically inferior compared to conventional techniques (con). This stems from the concern that the semicircular section leaves a triangular rim, which may preserve ramifications and potential complications. This nuanced addition has not only contributed to enhancing clarity but has also facilitated a more accessible comprehension of the study's objectives. Thank you for the suggestion!